# The Glenoid Track Concept: On-Track and Off-Track—A Narrative Review

Antonio Ventura [1,*], Claudia Smiraglio [2], Antonio Viscomi [2], Sergio De Salvatore [3] and Bernardo Bertucci [2]

1   Biocontrol SRL, 87100 Cosenza, Italy
2   Department of Radiology, Pugliese-Ciaccio Hospital, 88100 Catanzaro, Italy;
    claudiasmiraglio@yahoo.com (C.S.); pviscom@tin.it (A.V.); bernardo.bertucci118@gmail.com (B.B.)
3   Department of Orthopaedic and Trauma Surgery, Campus Bio-Medico University, 00128 Rome, Italy;
    s.desalvatore@unicampus.it
*   Correspondence: aventura@teletu.it

**Abstract:** Shoulder instability is described as a functional deficiency caused by excessive mobility of the humeral head over the glenoid. Various Glenohumeral Instability (GI) types have been described, but the traumatic anteroinferior form is the most frequent. The differences between engaging and non-engaging Hill–Sachs lesions (HSLs) are linked to bone loss assessment. On the contrary, the novel difference between "on-track" and "off-track" lesions is strictly related to surgical techniques. The specific involvement of glenoid and humerus bone defects in recurrent GI was poorly assessed in the literature before the glenoid-track concept (GT). Magnetic Resonance Imaging (MRI) and Arthro-MRI have been widely used to identify and characterize lesions to the ligamentous structures. However, only new technologies (3 Tesla MRI) accurately detect HSLs. On the contrary, Computed Tomography (CT) has been adopted to quantify glenoid bone deficit. The GT concept is a valuable tool for evaluating anterior shoulder instability in patients. Shoulders out of alignment may require more than just an arthroscopic Bankart, and a remplissage or bone transfer may be necessary. Specifically, isolated Bankart repair should be considered in patients with recurrent instability and an on-track lesion with less than 25% glenoid bone loss. In off-track lesions and less than 25% glenoid bone loss, remplissage should be used. Bone transplant surgery is required for patients with a glenoid bone defect of more than 25%. This narrative review aims to report the most updated findings on "on-track" and "off-track" lesions in GI.

**Keywords:** glenoid track; on-track; off-track; Hill–Sachs index; bone loss; engagement; instability

## 1. Introduction

Shoulder instability is described as a functional deficiency caused by excessive mobility of the humeral head over the glenoid [1]. Such instability could occur after trauma, overloads, or congenital laxity [2]. Glenohumeral Instability (GI) can be classified by etiology, severity, frequency, and direction of dislocation [3–5]. According to its etiology, GI could be divided into traumatic and atraumatic dislocations. Moreover, another cause of dislocation, named "micro instability", was found in overhead athletes [6,7]. Various GI types have been described, but the traumatic anteroinferior form is the most frequent [8]. One cause of anterior instability, particularly with the shoulder in abduction and external rotation, is the Hill–Sachs lesion (HSL) [9]. The humeral head is translated anteriorly, and the capsulolabral structures are injured. In addition, if there is an excessive displacement of the humeral head, a compression fracture may occur at the posterior and superolateral portions of the humeral head due to impact with the glenoid [10].

The differences between engaging and non-engaging HSLs are linked to bone loss assessment. On the contrary, the novel difference between "on-track" and "off-track" lesions is strictly related to the surgical techniques [11,12].

Before introducing the concept of the glenoid track (GT), several researchers examined the specific involvement of glenoid and humerus bone defects in recurrent GI [3,13]. However, only a few investigated the combined roles. The portion of the posterior humeral articular surface that encounters the glenoid when the arm moves over the posterior end-range of motion is the GT [12] (Figure 1). The glenoid width determines the GT's width (measured by CT scan). However, the GT was calculated as glenoid width x 0.84. A bone defect on the anterior edge of the glenoid determines a reduction in the GT width. Thus, in a glenoid bone defect, 84% of the glenoid width should be removed from the defect width.

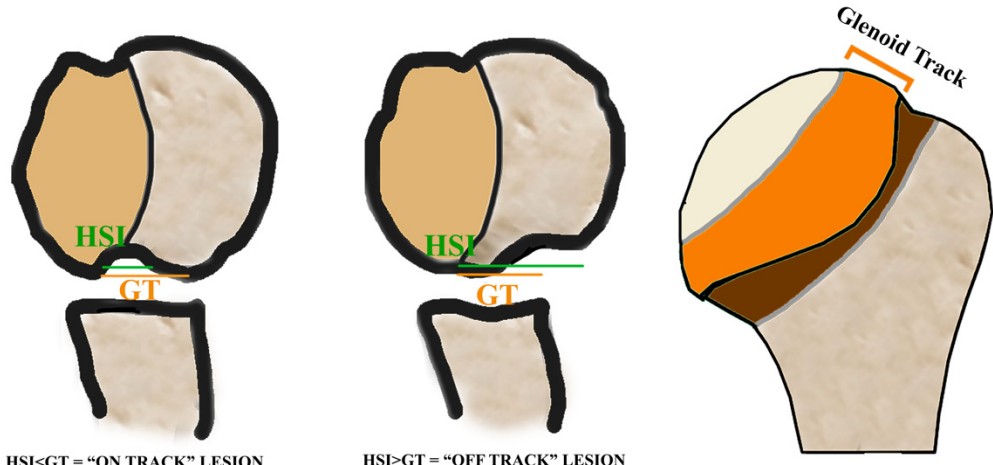

**Figure 1.** The glenoid track concept and "on-track and off-track lesions". HSI: Hill–Sachs index; GT: glenoid track.

In 2014, Di Giacomo et al. first integrated HSL quantification into the diagnostic and therapeutic algorithm of anterior instability [14]. The authors wanted to know if an HSL was on-track or off-track. Di Giacomo et al. devised a technique (combining radiographic and arthroscopic results) for determining whether an HSL would come into contact with the anterior glenoid rim, considering the possible presence of an anterior glenoid defect. An HSL that engages was named an "off-track" HSL. On the contrary, if it does not engage, it is called an "on-track" lesion [14]. The authors used the contralateral shoulder as a control to determine the magnitude of the glenoid defect. First, they measured the glenoid's maximum horizontal distance on both shoulders. Then, they calculated the defect size as the intact glenoid width minus the injured glenoid width. A line drawn at a distance of 83% of the glenoid width was set from the medial margin of the footprint. In the absence of a bony defect, this line represented the medial margin of the GT. On the contrary, in glenoid bony defect patients, the distance from the 83% line was subtracted to obtain the medial margin of the true GT (Figure 2).

The authors described a bone bridge still intact between the footprint and the lateral boundary of the HSL. The bone bridge width added to the HSL width was named the Hill–Sachs index (HSI). The critical point in identifying whether an HSL is on-track or off-track is the medial margin of the HSI (Figure 3).

Di Giacomo and colleagues concluded that HSLs located within the GT would be named "on-track HSLs" (Figure 4).

On the other hand, lesions that extend more medially over the medial margin of the GT would be defined as "off-track HSLs" (Figure 5) [14].

Recently, some authors have also emphasized the role of a bone deficit on the humerus, introducing the concept of bipolar bone loss [15]. X-rays, Computed Tomography (CT), and Magnetic Resonance Imaging (MRI) could be used to estimate the glenohumeral bone loss (uni- or bipolar), grade soft tissue lesions, and identify the postoperative risk of recurrence, being fundamental in the preoperative assessment of patients with traumatic GI [1,16,17].

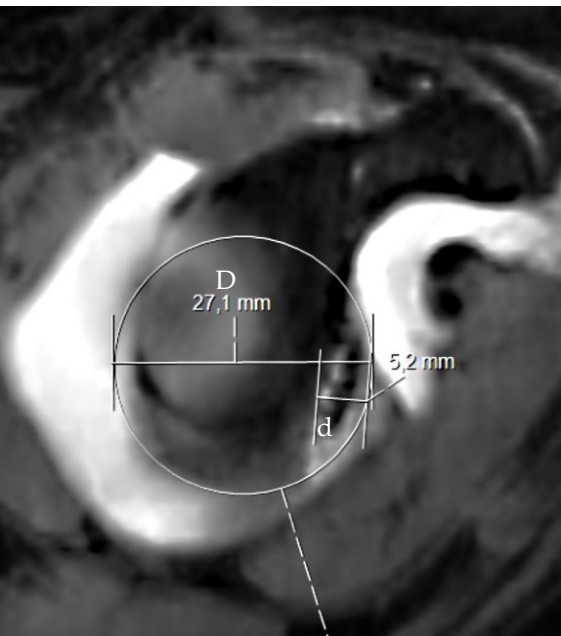

**Figure 2.** The glenoid track on MRI arthrography. A fat-saturated T1-weighted axial MRI arthrograph was obtained with a 1.5 T imager. A circle is tracked, including the posterior and inferior margins of the glenoid. The glenoid track was calculated as 0.83 (D − d). Here, D (27.1 mm) represents the diameter of the intact glenoid, while d (5.2 mm) is the amount of glenoid bone loss.

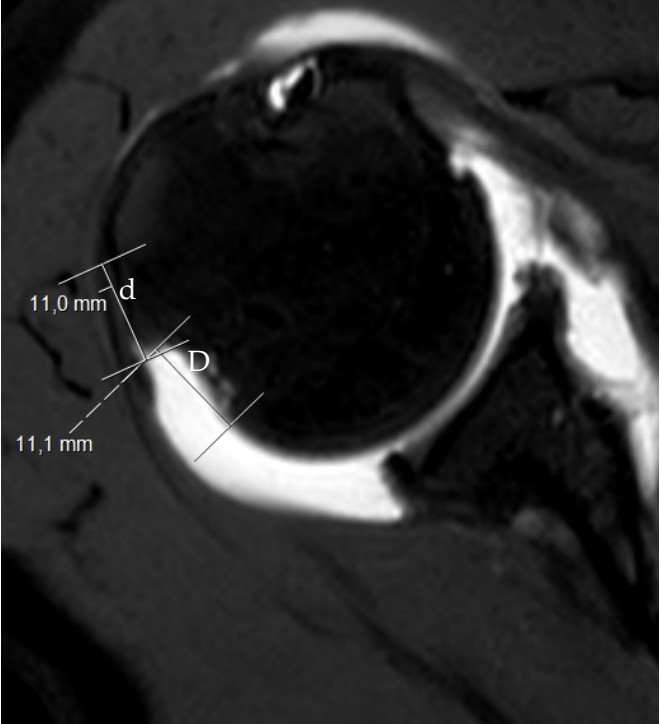

**Figure 3.** The Hill–Sachs index (HSI) on Magnetic Resonance Imaging (MRI) arthrography. The HSI was calculated as D + d. Here, D (11.1 mm) represents the width of the Hill–Sachs lesions (HSLs), while d (11.0 mm) is the width of the intact bone part between the footprint and the lateral margin of the HSL.

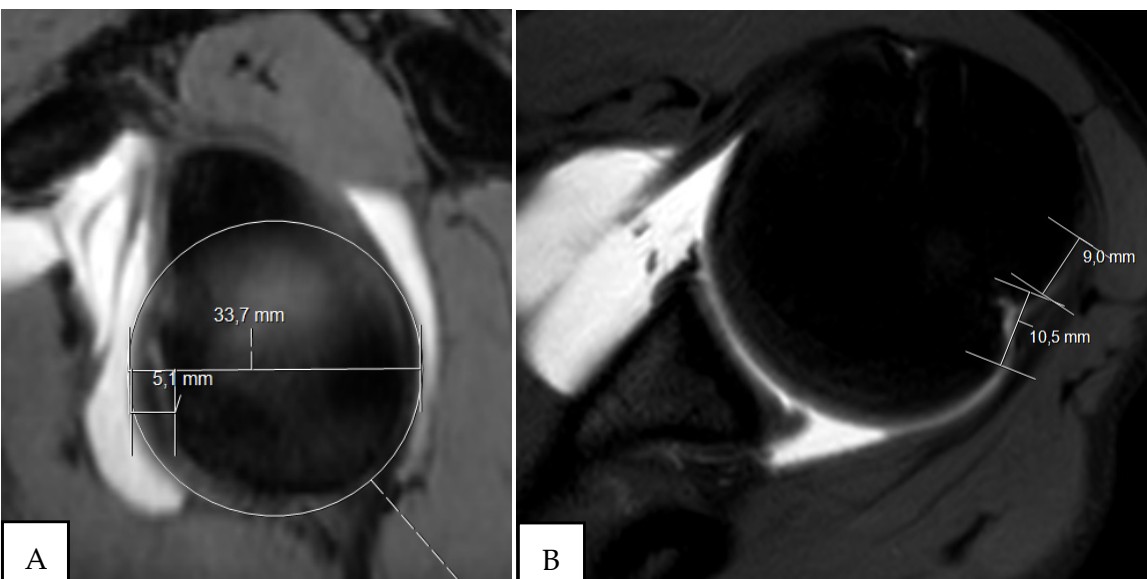

**Figure 4.** On-track lesion on Magnetic Resonance Imaging (MRI) arthrography. (**A**) In this case, the glenoid track is equal to 23 mm ($33 - 5 \times 0.83 = 23$). (**B**) The Hill–Sachs index (HSI) is 19.5 mm ($10.5 + 9$). It is an on-track lesion because the glenoid track (23 mm) is greater than the HSI (19.5 mm).

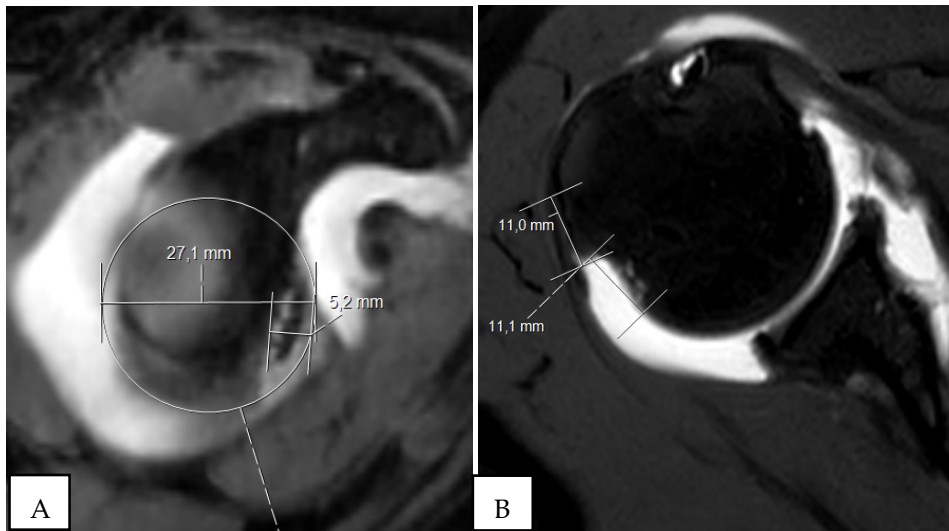

**Figure 5.** Off-track lesion on Magnetic Resonance Imaging (MRI) arthrography. (**A**) In this case, the glenoid track is equal to 18.17 mm ($27.1 - 5.2 \times 0.83 = 18.17$). (**B**) The Hill–Sachs index (HSI) is 22.1 mm ($11.1 + 11$). It is an off-track lesion because the glenoid track (18.17 mm) is less than the HSI (22.1 mm).

The GT concept is a helpful tool for assessing individuals with anterior shoulder instability. Shoulders out of alignment may require more than an arthroscopic Bankart, and a remplissage or bone transfer may be necessary [18].

This narrative review aims to report the most updated findings on "on-track" and "off-track" lesions in GI.

## 2. The Glenoid Track and Hill–Sachs Lesions: On-Track and Off-Track

Before the concept of bipolar lesions, capsule-ligamentous structure injuries and glenoid bone loss were considered the leading causes of anterior GI [19]. MRI and Arthro-MRI have been widely used to identify and characterize lesions to the ligamentous structures. However, the results on MRI's accuracy in detecting HSLs are controversial. Godinho

and colleagues reported that MRI and Arthro-MRI presented low accuracy in detecting HSLs [20]. On the contrary, according to Vopat and colleagues, CT has the greatest power in diagnosing and quantifying glenoid bone deficit [21].

However, new generations of MRI with 3 Tesla power could help identify soft tissue injuries, such as capsular and labral lesions [12]. Moreover, 3 Tesla MRIs could have a higher resolution compared to CT. Nowadays, thanks to the improvement in MRI technologies, Arthro-MRI can be used only in selected cases, given that it is a demanding procedure, and its accuracy ranges from 81% to 93% [21]. Therefore, MRI and Arthro-MRI could be used independently or in conjunction with CT scans.

After a Bankart repair without bone grafting, Itoi and colleagues found that a bony glenoid deficiency of at least 21% of the superior–inferior glenoid length caused instability [22,23]. Similarly, according to Burkhart and colleagues, a lesion spanning more than 25% of the glenoid breadth necessitates bone grafting [24]. Burkhart and De Beer examined the outcomes of 194 successive arthroscopic Bankart repairs, revealing specific characteristics that led to recurrences of instability. A bony defect of the humeral head that engages with the anterior glenoid rim during shoulder abduction and external rotation (ABER) represents one of the most important risk factors [25].

In 2007, Yamamoto and Itoi described the role of the contact zone between the glenoid and the humeral head in shoulder dislocations, defining it as the GT [23]. The authors measured the width of this area with the arm at various degrees of ABER. They demonstrated that with arm abduction, the glenoid contact zone shifts from the inferomedial to the superolateral part of the posterior articular surface of the humeral head, creating the so-called GT. Moreover, they reported that the GT width was 84% of the glenoid width [23]. With the introduction of the GT concept, they were able to predict the chance of an HSL colliding with the glenoid rim and whether or not there was a bone defect.

In particular, Itoi and colleagues reported that if an HSL is confined within the GT, there is no possibility that the lesion exceeds the glenoid border causing the engaging [23]. Conversely, if an HSL's medial edge is not within the GT, the risk of engaging is higher due to the humeral head that could pass the glenoid border.

Yamamoto and Itoi introduced a new concept in 2007 [23]. The authors assessed the importance of considering the presence of bony defects both in the humeral head and in the glenoid. According to the authors, despite the edge of the posterolateral articular cartilage of the humeral head being injured in an HSL, the rotator cuff footprint on the greater tuberosity is often entire [26]. Therefore, the latter may be a preferable metric to use when comparing the breadth of the GT to the width of an HSL.

In 2011 Omori et al. used MRI to measure the width of the GT with the arm at 90° abduction. The authors reported that the GT was 85% ± 12% of the glenoid width [27].

In 2015, Gyftopoulos et al., in a retrospective review, used MRI to predict the risk of engagement using the GH concept [28]. In addition, the authors assessed the glenoid bone loss using the best-fit circle method [29] and the engagement according to the on-track, off-track method. As a result, Gyftopoulos and colleagues reported for the first time that MRI could be used to assess HSLs and glenoid bone loss and that the GT method is an effective technique for predicting engagement with an accuracy of 84.2% [28].

The difference between engaging and non-engaging lesions proposed by Burkhart and De Beer [25] and the glenoid track proposed by Yamamoto and colleagues are entirely congruent [23]. They are complementary ideas because they assess how bipolar bone loss interacts with dynamic shoulder function. While the glenoid track can be assessed by arthroscopy or CT, the existence of an engaged Hill–Sachs lesion can be found during arthroscopy with the arm in ABER [14]. According to Itoi and Boileau, all bipolar bone lesions are engaging, as the engagement was necessary for the development of the Hill–Sachs lesion.

### 3. Clinical Implications of the Glenoid Track Method

Understanding the pathogenesis and the characteristics of the lesion could guide the surgeon in choosing the most appropriate treatment for the patient. The glenoid track method has been progressively used in preoperative planning and has been promoted as a routine pre-surgical evaluation of all candidates for arthroscopic anterior GH repair [12,30]. In a retrospective study of 100 patients who had arthroscopic Bankart repair, Locher et al. discovered that the odds ratio of off-track vs. on-track patients with recurrence was 8.3 [16]. In 57 patients with arthroscopic Bankart repair, Shaha et al. examined the likelihood of recurrence [31]. The on-track patients had an 8% recurrence rate, whereas the off-track patients had a 75% recurrence rate. The off-track idea has a positive predictive value of 75% for predicting recurrence [31]. Di Giacomo et al. proposed a surgical algorithm using the GT method. Preoperative MRI or arthroscopy can classify a patient into four groups based on the degree of glenoid bone loss or whether the HSL is on-track or off-track [14].

Isolated Bankart repair is appropriate when a patient has recurrent instability and an on-track lesion with less than 25% glenoid bone loss [32]. In case of an off-track lesion and less than 25% glenoid bone loss, remplissage should be considered the appropriate treatment [32]. Bone transplant surgery is required for patients with a glenoid bone defect of more than 25% [14]. A recent study evaluated clinical results following arthroscopic Bankart repair with selective remplissage treatment in patients with and without off-track lesions and found that this strategy was effective [33]. Patients with an off-track lesion and those with on-track lesions, treated by arthroscopic Bankart repair with selective remplissage, reported similar clinical results and recurrence rates. As a result, after Bankart repair with capsular plication, a selective remplissage surgery conducted to engage the humeral head identified during arthroscopy should be considered a therapy for off-track lesions.

### 4. Conclusions

One of the most critical risk factors for surgical failure or the recurrence after surgery of anterior shoulder dislocation is bipolar bone loss. Therefore, a correct preoperative evaluation is necessary before surgery by CT, MRI, and Arthro-MRI. In addition, the GT predicts the risk of humeral head engagement and probable dislocation based on loss of the glenoid and humeral head bone. Therefore, the GT concept can potentially help surgeons in the decision process. However, further systematic reviews are required to understand the topic entirely.

**Author Contributions:** Conceptualization, A.V. (Antonio Ventura) and C.S.; methodology, A.V. (Antonio Viscomi); software, C.S.; validation, B.B.; formal analysis, B.B.; investigation, A.V. (Antonio Ventura); resources, A.V. (Antonio Ventura); data curation, A.V. (Antonio Viscomi); writing—original draft preparation, A.V. (Antonio Ventura); writing—review and editing S.D.S.; visualization, S.D.S.; supervision, B.B.; project administration, S.D.S. All authors have read and agreed to the published version of the manuscript.

**Funding:** This research received no external funding.

**Institutional Review Board Statement:** Not applicable.

**Informed Consent Statement:** Not applicable.

**Conflicts of Interest:** The authors declare no conflict of interest.

### Abbreviations

| | |
|---|---|
| Abduction and external rotation | ABER |
| Glenohumeral Instability | GI |
| Glenoid track | GT |
| Hill–Sachs index | HSI |
| Hill–Sachs lesions | HSL |
| Magnetic Resonance Imaging | MRI |

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
