# Peer review of "The Glenoid Track Concept: On-Track and Off-Track—A Narrative Review"

_2673-4036, doi:10.3390/osteology2030015_

Round 1

Reviewer 1 Report

Dear Editor,

dear authors,

Thank you very much for the opportunity to review the manuscript entitled:

 The glenoid track concept: on-track and off-track”

which I have read with great interest. Please find my detailed comments below.

-Extensive revision by a native speaker is recommended

Abstract seems just like a copy of the introduction, please revise and point out the main steps of diagnosis and management of bone loss in GI. Also add recommendations when that is indicated.

-What type of study / review do you present? Please indicate in the headline and abstract

-Please describe the mechanism leading to hill sachs defects in the introduction

-Please explain abbreviations before citing. That’s important because it makes the text hard to follow. Avoid abbreviations in sub-headlines.

-GT is not explained well for a narrative review , maybe add a figure.

-Please give detailed information on the MRI diagnostic parameters that are common /recommended. Why not use CT since it’s a bony pathology. I think both seems appropriate but MRI should have a high resolution (3T). Is Artho-MRI really necessary in all cases?

-Please better explain how the definition of on and offtrack lesions was evaluated and which lesion will probably lead to higher recurrences of dislocations. Why are the engaging ones called off track? Those explanations should occur before the methods description..you do that well later in the text.. maybe move, makes the manuscript easier to understand

-Line 97: The medial margin of the footprint and the HSL was found. That sentence is not understandable in its context.

-Figure 1: quality is inferior, makes it hard to recognize the anatomical landmarks. Please provide higher resolution. Also indicate which type of MRT (tesla and sequence you would recommend). It also semms the figure only represents the Glenoid bone loss but does not demonstrate the GT concept.. maybe a drawing next to the MRI scabn would make that more readable.

Author Response

Dear reviewer,

We would like to thank you for the helpful comments and suggestions. We have revised the paper accordingly and hope that the work is now ready for publication. The changes are itemized below with our comments on the reviewer’s suggestions. Changes made in the text are highlighted in yellow in the original manuscript.

Reviewer #1:

Dear Editor,

dear authors, 

Thank you very much for the opportunity to review the manuscript entitled:   “The glenoid track concept: on-track and off-track” which I have read with great interest. Please find my detailed comments below.

  • Extensive revision by a native speaker is recommended

Thanks for the comment. A native English speaker revised the manuscript

  • Abstract seems just like a copy of the introduction, please revise and point out the main steps of diagnosis and management of bone loss in GI. Also add recommendations when that is indicated.

Thanks for the comment. The abstract has been fully revised according to the suggestions

  • What type of study / review do you present? Please indicate in the headline and abstract

Thanks for the comment. The title and the abstract have been modified accordingly

  • Please describe the mechanism leading to hill sachs defects in the introduction

Thanks for the comment. The introduction has been improved according to your suggestions

  • Please explain abbreviations before citing. That’s important because it makes the text hard to follow. Avoid abbreviations in sub-headlines.

Thanks for the comment. The abbreviations have been revised throughout the text. We also added an abbreviation section to improve the readability

  • GT is not explained well for a narrative review , maybe add a figure.

 Thanks for the comment. The introduction has been modified accordingly and a figure has been added

  • Please give detailed information on the MRI diagnostic parameters that are common /recommended. Why not use CT since it’s a bony pathology. I think both seems appropriate but MRI should have a high resolution (3T). Is Artho-MRI really necessary in all cases?

 Thanks for the comment. The paper has been improved with your suggestions.

“Results on MRI accuracy in detecting HSL are controversial. Godinho and colleagues reported that MRI and artrho-MRI presented low accuracy in detecting HSL [19]. On the contrary, according to Vopat and colleagues, CT has the most power in diagnosing and quantify glenoid bone deficit [20].

However, new generations of MRI with 3 Tesla power could help identify soft tissue injuries, such as capsular and labral lesions [12]. Moreover, 3 Tesla MRIs could have a high resolution compared to CT. Nowadays, thanks to the improvement in MRI technologies, arthro-MRI can be used only in selected cases, being a demanding procedure, and its accuracy ranges from 81% to 93% [20]. Therefore, MRI and arthro-MRI could be used independently or in conjunction with CT scans.”

  • Please better explain how the definition of on and offtrack lesions was evaluated and which lesion will probably lead to higher recurrences of dislocations. Why are the engaging ones called off track? Those explanations should occur before the methods description..you do that well later in the text.. maybe move, makes the manuscript easier to understand

 Thanks for the comment. The manuscript has been modified accordingly. The explanation of the on-off track is now moved to the introduction. In addition, the difference between engaging and non-engaging is reported now on page 6.

“The difference between engaging and non-engaging lesions proposed by Burkhart and De Beer [25] and the glenoid track proposed by Yamamoto and colleagues are both entirely congruent [23]. They are complementary ideas because they assess how bipolar bone loss interacts with dynamic shoulder function. While the glenoid track can be assessed by arthroscopy or CT, the existence of an engaged Hill-Sachs lesion can be found during arthroscopy with the arm in ABER [14]. According to Itoi and Boileau, all bipolar bone lesions are engaging, as the engagement was necessary for the development of the Hill-Sachs lesion.”

  • Line 97: The medial margin of the footprint and the HSL was found. That sentence is not understandable in its context.

Thanks for the comment. The sentence has been deleted

  • Figure 1: quality is inferior, making it hard to recognize the anatomical landmarks. Please provide higher resolution. Also indicate which type of MRT (tesla and sequence you would recommend). It also semms the figure only represents the Glenoid bone loss but does not demonstrate the GT concept.. maybe a drawing next to the MRI scan would make that more readable.

Thanks for the comment. Thanks to your previous comment we added a new figure 1 focused on the glenoid concept. We improved the figure caption with the information required. Unfortunately, this is the maximum resolution we could provide. In addition, we improved the quality of the imaging using upscaling software.

Reviewer 2 Report

Authors presented a manuscript on the summary of factors contributing shoulder (glenohumeral) instability in the current literature, and proposed the concept of glenoid on- and off- track. It is suggested that author may consider adding 'a narrative review' in the title for highlight.

The Abstract section may need to be rewritten, as authors kept using 'several researchers' and 'some authors'. The authors seems repeating what is already existed in the literature. The abstract should be a summary of what is this manuscript about.

Authors may consider adding an independent 'methodology' section to indicate what strategy was employed to search the literature, and the criteria of literature inclusion and exclusion. 

Authors may add a part introducing the practical and clinical implications of this study, as take-home message for readers.

Author Response

Dear reviewer,

We would like to thank you for the helpful comments and suggestions. We have revised the paper accordingly and hope that the work is now ready for publication. The changes are itemized below with our comments on the reviewer’s suggestions. Changes made in the text are highlighted in yellow in the original manuscript.

Reviewer 2

  • Authors presented a manuscript on the summary of factors contributing shoulder (glenohumeral) instability in the current literature, and proposed the concept of glenoid on- and off- track. It is suggested that author may consider adding 'a narrative review' in the title for highlight.

Thanks for the comment. We are honoured you appreciated our paper. The title has been modified accordingly.

  • The Abstract section may need to be rewritten, as authors kept using 'several researchers' and 'some authors'. The authors seems repeating what is already existed in the literature. The abstract should be a summary of what is this manuscript about.

Thanks for the comment. The abstract has been revised according to the reviewer’s suggestions

  • Authors may consider adding an independent 'methodology' section to indicate what strategy was employed to search the literature, and the criteria of literature inclusion and exclusion. 

Thank you for giving us the possibility to explain this point better. As the present study is a narrative review and not a systematic review, no codified strategy was adopted to include the articles. This study only aims to inform clinicians on this topic. Further systematic reviews are required to understand the topic better. Thanks to your comment, we improved the conclusion with this limitation.

  • Authors may add a part introducing the practical and clinical implications of this study, as a take-home message for readers.

Thanks for the comment. The clinical implication of this study is reported in page 7. We modified the title of the headings to improve the readability and focus on the study's practical and clinical implications.

Round 2

Reviewer 1 Report

All comments have been adressed properly. The manuscript has improved and can now be recommended for publication.

I would like to thank the authors for their effort.

Reviewer 2 Report

Authors now addressed the concerns and comments for the previous manuscript. The quality is improved substantially.